# Delayed Stroke after Aneurysm Treatment with Flow Diverters in Small Cerebral Vessels: A Potentially Critical Complication Caused by Subacute Vasospasm

**DOI:** 10.3390/jcm8101649

**Published:** 2019-10-10

**Authors:** Stefan Schob, Cindy Richter, Cordula Scherlach, Dirk Lindner, Uwe Planitzer, Gordian Hamerla, Svitlana Ziganshyna, Robert Werdehausen, Manuel Florian Struck, Bernd Schob, Khaled Gaber, Jürgen Meixensberger, Karl-Titus Hoffmann, Ulf Quäschling

**Affiliations:** 1Department of Neuroradiology, University Hospital Leipzig, 04103 Leipzig, Germany; Cindy.Richter@medizin.uni-leipzig.de (C.R.); Cordula.Scherlach@Medizin.uni-Leipzig.de (C.S.); gordian.hamerla@Medizin.uni-Leipzig.de (G.H.); Karl-Titus.hoffmann@medizin.uni-leipzig.de (K.-T.H.); Ulf.quaeschling@medizin.uni-Leipzig.de (U.Q.); 2Department of Neurosurgery, University Hospital Leipzig, 04103 Leipzig, Germany; Dirk.Lindner@medizin.uni-leipzig.de (D.L.); Uwe.Planitzer@Medizin.uni-Leipzig.de (U.P.); Khaled.Gaber@medizin.uni-leipzig.de (K.G.); juergen.meixensberger@medizin.uni-leipzig.de (J.M.); 3Department of Anaesthesiology, University Hospital Leipzig, 04103 Leipzig, Germany; Svitlana.Ziganshyna@medizin.uni-leipzig.de (S.Z.); Robert.Werdehausen@Medizin.uni-Leipzig.de (R.W.); ManuelFlorian.struck@medizin.uni-leipzig.de (M.F.S.); 4Department for Lightweight Structures and Polymers, Technical University Chemnitz, 09126 Chemnitz, Germany; Bernd.Schob@web.de

**Keywords:** flow diversion, delayed ischemia, stroke, vasospasm, hemodynamic therapy, cerebral aneurysm

## Abstract

Flow diversion (FD) is a novel endovascular technique based on the profound alteration of cerebrovascular hemodynamics, which emerged as a promising minimally invasive therapy for intracranial aneurysms. However, delayed post-procedural stroke remains an unexplained concern. A consistent follow-up-regimen has not yet been defined, but is required urgently to clarify the underlying cause of delayed ischemia. In the last two years, 223 patients were treated with six different FD devices in our center. We identified subacute, FD-induced segmental vasospasm (SV) in 36 patients as a yet unknown, delayed-type reaction potentially compromising brain perfusion to a critical level. Furthermore, 86% of all patients revealed significant SV approximately four weeks after treatment. In addition, 56% had SV with 25% stenosis, and 80% had additional neointimal hyperplasia. Only 13% exhibited SV-related high-grade stenosis. One of those suffered stroke due to prolonged SV, requiring neurocritical care and repeated intra-arterial (i.a.) biochemical angioplasty for seven days to prevent territorial infarction. Five patients suffered newly manifested, transient hemicrania accompanying a compensatorily increased ipsilateral leptomeningeal perfusion. One treated vessel obliterated permanently. Hence, FD-induced SV is a frequent vascular reaction after FD treatment, potentially causing symptomatic ischemia or even stroke, approximately one month post procedure. A specifically early follow-up-strategy must be applied to identify patients at risk for ischemia, requiring intensified monitoring and potentially anti-vasospastic treatment.

## 1. Introduction

Hemodynamic treatment of cerebral aneurysms employing flow-diverting stents (FDS) emerged as an endovascular approach of at least equal clinical importance compared to conventional techniques like coiling and clipping, although its application remains limited to proximal, rather large segments of cerebral vessels [1,2]. FDS consist of densely woven alloy meshes, achieving a distinctly increased endovascular surface coverage compared to customary aneurysm stents, thus affecting the cerebral hemodynamic profoundly. More specifically, after implantation, the FDS immediately reduces aneurysmal influx and re-adjusts blood flow along the physiological axis of the parent vessel. After incremental aneurysmal thrombosis and formation of a neointima along the FDS mesh, the aneurysm is excluded from intracranial circulation [3]. Although excellent occlusion rates of even challenging aneurysms accompanied by comparatively low procedural complication rates were reported, pathophysiologically yet obscure, delayed ischemic adverse events, despite sufficient platelet function inhibition, remain a concern [4,5,6]. In this regard, FDS-induced hemodynamic insufficiency with subsequent watershed ischemia, neointimal hyperplasia, in-stent stenosis, and secondary device deformations (fish-mouthing, peripheral tapering) were suspected as elicitors in some cases [7,8,9,10,11].

Consistent guidelines regarding the time point and appropriate modality for follow-ups after FDS implantation are not yet defined [12]. Therefore, as it is generally accepted for endovascular aneurysm treatment, the first time point for follow-up imaging in our neurovascular center was set at six months post intervention [12,13]. Related to the recent limited marketing release of a low-profile FDS, specifically developed for the unprecedented treatment of small cerebral vessels [14], we defined a follow-up regimen, henceforth including imaging at three, nine, and 24 months after the intervention. 

After experiencing a yet unique, neurologically symptomatic case of severe vasospasm beginning three weeks post implantation, which was strictly confined to the FDS stent bearing segment and caused significant FDS compression culminating in long-segmental high-grade stenosis, we decided to immediately include very early device imaging in all further cases. This supplementary follow-up three–four weeks post procedure included plain device radiography as a first step, with the aim to screen for device compressions potentially causing critical stenosis in all FDS-treated patients, and a combination of digital subtraction angiography (DSA) and magnetic resonance imaging (MRI) as a further diagnostic escalation in the case of severe stenosis. Especially considering previous reports on delayed type, although probably mostly clinically inapparent FDS-associated post-ischemic lesions [4,5], we initiated this investigation of early and very early follow-ups after FDS implantation in our department, specifically looking for characteristic FDS compressions suggesting FDS-induced, subacute vasospasm. 

## 2. Material and Methods

### 2.1. Ethics Approval

Our study, investigating cases from May 2017 to July 2019, was approved by the institutional ethics committee (local IRB nr. AZ 208-15-0010062015). Informed consent of each patient regarding the scientific use of radiological and clinical data was obtained in writing either from the patient or their legal representative.

### 2.2. Patients, Quantification of Vasospastic Deformities, Intimal Hyperplasia, and Cumulative Stenosis

FDS implantations in the last two years providing (yet unconventional) early follow-up investigations ≤6 months post intervention were included. A total of 223 cases of FDS implantations were identified; however, only in 36 cases (15 male, 21 female patients; mean age of 42 years, ranging from 18 to 78 years) was follow-up imaging 1–5 months post intervention performed. 

Two of the cases were treated for acute aneurysmal subarachnoid hemorrhage (SAH) using FDS as a primary strategy. Twenty-three cases received FDS treatment in the course of the plug and pipe algorithm (coiling first, followed by FDS for treatment completion). Twelve incidentally detected aneurysms were treated using FDS as a solitary initial strategy. Three cases of FDS implantation were performed within a previously stented segment for completion of eventually insufficiently treated aneurysms (2 × “elective” plug and pipe (P&P), 1 × P&P after initial SAH). Demographic data, location, size and morphology, immediate clinical and early angiographic follow-up results, and complications were recorded. An overview of individual clinical and demographic data is provided in Table 1.

The degree of vasospastic deformities was evaluated employing a modified version of the North American Symptomatic Carotid Endarterectomy Trial—‘NASCET’ calculation (1 − (diameter of the most stenotic segment/corresponding original diameter measured immediately after implantation) × 100). Neointimal hyperplasia was defined as the non-opacified distance between endoluminal contrast and the inner border of the flow diverter (exemplarily demonstrated in Figure 2). Cumulative stenosis was calculated as vasospastic stenosis, in addition to narrowing due to neointimal hyperplasia.

### 2.3. Interventional Procedures

Informed consent for elective endovascular treatment was obtained from all patients. Oral dual platelet medication (500 mg of acetylsalicylic acid (ASA) and 180 mg of ticagrelor) was started the day before the procedure in the elective cases. On the day of the procedure, the standard regimen, consisting of 100 mg of ASA (once a day, life-long) and 90 mg of ticagrelor (twice a day, for 12 months), was initiated. 

In the case of emergency treatments, intravenous dual platelet inhibition was initiated after interdisciplinary consent was gained for FDS implantation. For this purpose, 500 mg of ASA and 180 µg of eptifibatide/kg bodyweight were given prior to intervention, followed by oral administration of 180 mg of ticagrelor after the procedure. The abovementioned standard regimen was continued accordingly.

All endovascular procedures were performed under general anesthesia using a bi-planar angiography suite. Endovascular access was established via the right femoral artery using an 8-French introducer sheath. A bolus of heparin (5000 units) was administered via the sheath initially. All procedures were performed by two each of three available neuro-interventionalists with five, 14, and 18 years of experience. 

Biochemical angioplasties were performed only in the symptomatic case of severe vasospasm. For this, a 4F diagnostic catheter (4F vertebral, cordis, FL, USA) was placed in the distal extracranial internal carotid artery (ICA). In order to relieve vasospastic stenosis, nimodipine, milrinone, and alprostadil were applied intra-arterially (i.a.) in sequential order, which increased cerebral perfusion significantly.

### 2.4. Statistical Analysis

Statistical evaluation was performed using SPSS 24.0 (IBM, Armonk, NY, USA). Normal distributions of vasoconstrictive deformities, intimal hyperplasia, and cumulative stenosis were evaluated using the Shapiro–Wilk test accounting for the following items as differentiating variables: Gender;Localization of the aneurysms;Severity of disease in case of initial SAH;Status of the aneurysm—unruptured vs. ruptured;Endovascular strategy (FDS only for incidental aneurysm treatment; FDS only for acutely ruptured aneurysm treatment; plug and pipe: coiling in acute SAH and subsequent FDS implantation; complementary FDS after initial stent-assisted coiling: FDS in-stent implantation).

As a second step, analysis of variance (ANOVA) and the Kruskal–Wallis test were calculated to identify significant differences regarding vasoconstrictive deformities, intimal hyperplasia, and total stenosis after forming subgroups according to the aforementioned differentiating variables. ANOVA was employed in the case of normal distribution, and Kruskal–Wallis was performed for non-normally distributed variables.

In a last step, correlation analysis was performed using Spearman’s rho coefficient to investigate the empirically suggestive relation between oversizing and delayed vasoconstrictive deformities, intimal hyperplasia, and cumulative stenosis. Also, the correlation between patient age and vasoconstrictive deformities was evaluated.

## 3. Results

### 3.1. Patients, Clinically Relevant Features, and Implanted Flow Diverter Stents

Finally, 36 patients (21 females, 15 males) were included in our evaluation. In this group, 26 patients received a Silk Vista Baby FDS (SVB, Balt, France), two were treated with a Pipeline Embolization Device 2 FDS (PED2, Medtronic, MN, USA), seven had a p48 FDS (Phenox, Germany), and one had a p64 implanted (Phenox, Germany). Table 1 provides detailed information on all cases included. Table 2 shows summarized data after grouping patients according to their degree of vasospastic deformations.

Two patients were treated with FDS as a first approach in acute SAH. Fourteen patients were treated with FDS only for incidental aneurysms. Eighteen patients received coiling only in the acute SAH setting with FDS implantation after recovery from the hemorrhagic event (plug and pipe). Four patients were treated with FDS implantation after stent-assisted coiling (SAC). In two cases, the FDS overlapped with the distal end of the previously implanted aneurysm stent (pCONus, Phenox, Germany); in the remaining cases, the FDS overlapped with the proximal end of the previously implanted device (LEO, Balt, France). 

Hunt and Hess grades, as well as Fisher scores, in the patients with initially ruptured aneurysms were as follows: Hunt and Hess, 2: *n* = 11; Hunt and Hess, 3: *n* = 8; Hunt and Hess, 4: *n* = 1; Fisher 2, *n* = 2; Fisher 3, *n* = 17; Fisher 4, *n* = 1.

### 3.2. Follow-Up Imaging

In 27 patients, follow-up imaging, providing endovascular contrast and information on device morphology, was available. In these cases, subacute segmental vasospasm and neointimal hyperplasia were rated individually and subsequently calculated as cumulative stenosis (FDS compression + intimal hyperplasia). In nine patients, radiography only (without endovascular application of contrast medium) was performed as the earliest follow-up modality. In these cases, only segmental vasospasm (not neointimal hyperplasia) was assessable. In 29 patients, very early follow-up imaging (<15 weeks) was available, whereas seven patients had imaging between 16 and 28 weeks after implantation.

### 3.3. Clinical Outcome

Only one of 36 patients (patient 27) experienced a severe, enduring symptomatic vasospasm (Figure 1), culminating in transiently symptomatic stroke three weeks after an uneventful procedure. The severe course of vasospasm required daily repeated i.a. treatments to enhance hemispheric perfusion and finally prevent impending territorial infarction. Details of the clinical course and hemodynamic adjustment under intensified neurocritical care, including eventually successful anti-vasospastic treatment for approximately one week, are provided below. 

One patient suffered spasm-related occlusion of the treated, left-sided dominant vertebral artery five months after FDS implantation. High-grade stenosis due to subacute vasospasm and neointimal hyperplasia appeared five weeks post implantation (Figure 2). However, the patient remained neurologically asymptomatic at all times, except for occasional episodes of temporary, diffuse, occipito-cervical pain. 

Five additional patients (9, 18, 26, 32, and 35) also complained of transient, atypical, unprecedented migraine-like headaches, occurring between three and five weeks post procedure, manifesting unihemispheric ipsilaterally to the treated vessel (for example, Figure 3). The headache disappeared on average two weeks after onset and responded well to common pain medication. 

Otherwise, no transient or persisting neurological deficits were detectable. All remaining patients presented healthy without any neurological (or otherwise procedure-related) complaints to our outpatient clinic and follow-up imaging.

### 3.4. Presentation and Further Course of the Patient Suffering from Stroke Due to Severe FDS-Induced Vasospasm

The patient was treated uneventfully with an FDS for a recurrent, initially ruptured aneurysm of the left internal carotid artery bifurcation, following the plug and pipe strategy (Figure 4).

After an unremarkable period of three weeks, the patient presented to our outpatient clinic suffering from fluctuating global aphasia and severe migraine-like headache. Lab tests validated sufficient efficacy of dual platelet inhibition (1 × 100 mg of ASA + 2 × 90 mg of ticagrelor orally per day), and the patient confirmed the regular, uninterrupted intake of her medication. Cranial MRI was performed immediately (Figure 5) and revealed a novel, small, T_2_-Fluid Attenuated Inversion Recovery(FLAIR)-weighted hyperintense lesion in the left-sided distal middle cerebral artery (MCA) territory, consistent with a small subcortical infarct. Furthermore, FLAIR imaging showed distinct hyperintensity of the peripheral segments of the affected MCA, indicating decreased blood flow in the MCA; however, metal artefacts impeded precise assessment of the stent-bearing vessel in time-of-flight (TOF) angiography. Figure 5 provides corresponding MRI findings in the transiently aphasic patient. 

Thereupon, the patient was transferred immediately to the angiography suite for further evaluation. Digital subtraction angiography (DSA) (Figure 6, middle row) showed severe subacute vasospasm with compression of the proximal and distal portions of the FDS, resulting in high-grade stenosis of the terminal intracranial ICA and the proximal M1 segment. Interestingly, the middle portion of the FDS, possessing the greatest extent of radial force [15], remained unaffected. As a consequence of the morphologically distinct, vasospastic tandem stenosis, left-hemispheric perfusion was reduced to less than 50% of the MCA territory, whilst the ipsilateral anterior cerebral artery (ACA) and the adjacent border zone parenchyma were now supplied by the right ICA via the AcomA and (retrogradely perfused, FDS-covered) left A1 segment (Figure 6, middle row). 

Informed patient consent and interdisciplinary consent were achieved for immediate selective biochemical angioplasty. The procedure markedly decreased device deformation and subsequent tandem stenosis. Left-hemispheric perfusion was restored and almost reached the pre-interventional status. The patient was admitted to our neuro-intensive care unit for close monitoring. During the stay on the neuro-intensive care unit, the patient was tested conclusively negative for a variety of possibly vasospasmogenic conditions: recurrent SAH, allergic reaction to the implanted device, vasculitis, and other rheumatic or infectious systemic inflammatory and autoimmune conditions. Daily oral anti-vasospasm medication was initiated according to our regular regimen for vasospasm prevention after acute SAH, consisting of 3 × 20 mg of nimodipine orally per day.

However, subacute segmental vasospasm reoccurred for the following seven days, repeatedly causing high-grade tandem stenosis of the terminal intracranial ICA and proximal MCA. Daily transcranial ultrasounds were performed for early detection of recurrent vasospasms, considering peak velocity values of >160 cm/s in the MCA as a threshold for significant vasospasm. Ultrasound imaging duplex sonography revealed recurrent peak systolic velocities >220 cm/s every 10–12 h as a sign of vasospasm reoccurrence. The patient was transferred to the angio-suite for verification and subsequent mean arterial pressure-driven intra-arterial biochemical angioplasty. In every session, DSAs revealed severely impeded anterograde blood flow via the left ICA, as well as severely hindered compensatory collateral blood flow from the right ICA (via AcomA and the FDS-covered, left-sided A1 segment), necessitating selective intra-arterial biochemical angioplasty twice a day for a period of seven days. Intra-arterial spasmolysis was performed according our established SAH vasospasm protocol, which allows for three subsequent treatment steps. As a first step a combination of milrinone and nimodipine is slowly injected via the selectively probed, angiographically most affected intracranial vessel. Control angiograms are performed every 5–15 min whilst the patient is closely monitored (blood pressure—RR, heartrate, saturation, and electrocardiogram—ECG). If necessary, additional nitroglycerin is administered i.a. to augment vaso-relaxation. In cases of extraordinarily severe, comparatively resistant vasospasm, sildenafil was additionally injected. Figure 6 demonstrates the pre-implantation status (upper row), perfusion under left-sided segmental vasospasm (middle row), and final imaging after seven days of repeated selective i.a. spasmolysis (inferior row).

Eventually, the extent of subacute vasospasm decreased beginning on day six, allowing for a reduction of intra-arterial treatment on day seven and cessation of invasive biochemical angioplasty on day eight. The patient was closely monitored on the intensive care unit (ICU) for the four following days, including transcranial ultrasound examinations every 12 h. Flow velocities <140 cm/s were recorded during the four observational days on ICU. The patient completely recovered from her aphasia and did not show any residual neurological deficit. After a final thorough neurological examination, the patient was dismissed from our hospital, and oral anti-vasospasm medication was continued for three months. A follow-up MRI three months after discharge from our hospital revealed no additional FLAIR hyperintense lesions in the brain parenchyma or signs of decreased flow in the previously affected left ICA–MCA.

### 3.5. Imaging Findings in the Overall Patient Collective

Totally, 86% (31/36 cases) showed distinct, concentric deformations of the implanted FDS. A sum of 56% (20/36 cases) revealed concentric deformations causing more than 25% reduction in diameter of the stent-bearing segmental lumen. Eighty percent (16/20 patients) showed additional neointimal hyperplasia. Only 13% (5/36 cases) exhibited deformations causing more than 50% reduction in diameter of the stent-bearing segment. 

One dominant V4 segment (Figure 2) was finally occluded as a consequence of subacute vasospasm and neointimal hyperplasia, which was fully compensated by collateral flow without manifesting an infarction. Solitary intima hyperplasia was detectable in 8% (3/36 cases). 

Most frequently, subacute vasospasm with deformation of the implanted FDS occurred along the segment corresponding to the proximal landing zone: 15 cases (42%). Subacute vasospasm with deformations corresponding to the location of the distal landing zone occurred in nine cases (25%). Deformations corresponding to both landing zones occurred in nine cases (25%). Additional deformation in the center of the implanted device occurred in only two cases. Five cases (14%) showed no deformations at all. In the three cases of complementary FDS implantation, where the FDS was implanted into a previously stented segment, no deformities occurred along the stent-covered segment. However, the “unprotected” landing zone of the eventually implanted FDS exhibited characteristic deformities in all cases.

### 3.6. Statistical Findings

#### 3.6.1. Normal vs. Non Normal Distribution

Firstly, Shapiro–Wilk testing revealed non-normal distribution for subacute vasospasm with deformation values and neointimal hyperplasia extent, considering gender as a discriminating variable (each *p* < 0.05). Secondly, Shapiro–Wilk testing revealed non-normal distribution for vasoconstrictive deformation values and neointimal hyperplasia extent, considering the implanted device as a discriminating variable (each *p* < 0.05). Furthermore, Shapiro–Wilk testing revealed non-normal distribution only for neointimal hyperplasia extent, considering aneurysm location as a discriminating variable (*p* < 0.01). Values of vasoconstrictive deformation and cumulative stenosis were normally distributed among the locations. Using aneurysm status (ruptured vs. unruptured) as a discriminating variable, Shapiro–Wilk testing showed normal distributions for vasospasm, neointimal hyperplasia, and cumulative stenosis. Additionally, using the Hunt and Hess grading system and the Fisher score as discriminating variables, non-Gaussian distribution was revealed for vasospastic deformations, intimal hyperplasia, and cumulative stenosis (*p* < 0.05 each). Finally, considering the endovascular strategy as a discriminating variable, non-normal distribution was revealed for neointimal hyperplasia only (*p* < 0.01). Values of vasoconstrictive deformation and cumulative stenosis were normally distributed among the different endovascular strategies.

#### 3.6.2. Differences Concerning Vasospastic Deformation, Intimal Hyperplasia, and Cumulative Stenosis Comparing Relevant Subgroups

After grouping according to the aforementioned criteria (gender, FDS device, localization of the aneurysm, endovascular strategy, initial Hunt and Hess grade, and initial Fisher score), Kruskal–Wallis testing revealed the following results for non-normally distributed items:There were no significant differences in values of subacute vasospasm with stent deformities (*p* = 0.278), neointimal hyperplasia (*p* = 0.618), and cumulative stenosis (*p* = 0.443) when comparing male and female patients.The extents of neointimal hyperplasia did not differ significantly among the different aneurysm locations according to Kruskal–Wallis testing (*p* = 0.338).The extents of neointimal hyperplasia did not differ significantly between the distinct endovascular strategies according to Kruskal–Wallis testing (*p* = 0.116).There were no significant differences in values of subacute vasospasm with stent deformities (*p* = 0.151), neointimal hyperplasia (*p* = 0.201), and cumulative stenosis (*p* = 0.223) after using the initial Hunt and Hess grade as a grouping variable.The extents of subacute vasospasm, neointimal hyperplasia, and cumulative stenosis did not differ significantly among the groups after using Fisher score as a grouping variable (*p* = 0.331).

After grouping according to the aforementioned criteria (gender, FDS device, localization of the aneurysm, ruptured vs. unruptured aneurysm status, and endovascular strategy), ANOVA revealed the following results for normally distributed items:The extent of subacute vasospasm differed significantly among the different aneurysm locations (*p* = 0.007). More specifically, according Tukey’s post hoc analysis, vasoconstrictive deformities were significantly stronger at the ICA–MCA junction compared to the ICA–PcomA complex (*p* = 0.007), the MCA (*p* = 0.031), and the VBA (*p* = 0.018), whereas there was only a non-significant tendency of stronger vasoconstrictive deformities when comparing ICA–MCA junction and ICA–AcomA complex (*p* = 0.095).The extents of cumulative stenoses did not differ significantly between the distinct aneurysm locations, although a deformity-like tendency was observable (*p* = 0.089).The extents of vasoconstrictive deformities and cumulative stenoses did not differ significantly among the distinct endovascular strategies (*p* = 0.732 and *p* = 0.989).The extents of vasoconstrictive deformities, neointimal hyperplasia, and cumulative stenoses did not differ significantly between initially ruptured and incidental aneurysms (*p* = 0.821, *p* = 0.778, and *p* = 0.909).

Finally, Spearman’s rho calculation revealed moderate, statistically significant correlations as follows: device oversizing and subacute vasospasm (*R* = 0.379, *p* = 0.027); device oversizing and cumulative stenosis (*R* = 0.502, *p*
**=** 0.011). Spearman’s rho calculation revealed a similar tendency for device oversizing and neointimal hyperplasia; however, the correlation did not achieve statistical significance (*R* = 0.390, *p* = 0.054). Furthermore, correlative analysis revealed no significant correlation between age and subacute vasospasm (*R* = −0.246, *p* = 0.161).

As the occurrence of vasospasm might show a relation to the size of the individual aneurysm, groups were formed according the extent of vasospasm-related stenosis. The group showing 0–24% stenosing deformities had an average aneurysm size of 7.5 mm, ranging from 4.1 mm to 22 mm. The group showing 25–50% stenosing deformities had an average aneurysm size of 5.7 mm, ranging from <1 mm to 28 mm. The group showing the most severe deformities (>50%) had an average aneurysm size of 5.6 mm, ranging from 3.2 mm to 6.5 mm. 

## 4. Discussion

To the best of our knowledge, this study is the first reporting subacute, device-induced vasospasm as a frequent finding and potential cause for critical perfusion reduction in patients treated with different FDS. 

Interestingly, in patients with a previous episode of acute SAH, the severity of the initial hemorrhagic event (reflected by the Fisher scale or the Hunt and Hess scale) did not show an association to the extent of subacute, device-induced vasospasm. Furthermore, there was no significant difference in the severity of delayed, FDS-induced vasospasms when comparing acutely ruptured and unruptured aneurysms. Also, the size of the aneurysm did not vary to a relevant extent between the patients exhibiting non-mild versus mild to moderate or severe vasospasms.

Our results suggest a relationship between acquiescent oversizing of FDS and provocation of subacute segmental vasospasm. In particular, the vessel segments covering and being closely adjacent to the proximal and distal landing zones of the oversized FDS exhibited significant de novo narrowing. It is important to emphasize that the device-induced vasospasm was almost exclusively visible in early follow-up investigations, empirically peaking approximately one month after implantation, and it remained sufficiently recognizable until three, and, in very few cases, six months after implantation. The extent of segmental vasospasm with compression in the FDS was significantly associated with the maximum difference in diameter between target vessel and FDS. Vice versa, precise sizing or discrete undersizing did not result in significant vasospasm. Also, earlier implanted stents, which covered only one landing zone of a subsequently implanted FDS, did prohibit any vasoconstrictive reaction in this particular zone along the pre-existing stent. In contrast, the “unprotected” landing zone of the more recently placed FDS always showed significant vasospasm. This peculiarity is certainly related to a decreased effective radial force of the FDS, caused by the mechanically protective stent which was implanted in the earlier intervention.

Interestingly, mostly the proximal and distal portions of the FDS-associated segments had significant vasospasm. The central portion mostly remained unaffected or at least only discretely compressed. We consider this phenomenon to be a consequence of the previously reported differences in radial force when comparing the central portion of a stent and its more peripheral sections [15]. More specifically, we hypothesize that, in the majority of cases, the constrictive force of the arterial wall is exceeded by the radial outward force of the central part of the FDS. However, corresponding to the distally directed negative gradient of each device’s radial force, the ends of the device succumb to the muscular force of the circumferentially intact arterial wall. This hypothesis is corroborated by the fact that FDS-associated vasospasm exclusively occurred in areas of a fully developed and intact arterial wall, but was absent in the structurally weaker aneurysmatic segment. A potentially additional impact of the specific design of terminal segments of the device with more or less micro-traumatic effects of flared ends on the exposed vessel layers may additionally contribute to the preferred site of vasoconstriction.

Reviewing the literature, we found a number of studies reporting occurrences of comparable stent-related, segmental vasospasms in a variety of extra-cranial arteries [16,17,18], as well as cerebral vessels [10,11,19,20,21,22,23,24]. Considering those reports and our findings in the context of recent studies reporting the yet unexplainably high incidence of delayed type, non-procedure-associated ischemic lesions after flow diverter implantation [4,5], we postulate that device-induced segmental vasospasm plays a significant role in the genesis of these lesions.

Recently, a comparable but less striking association between stent oversizing, employing conventional aneurysm stents with significantly lower endovascular surface coverage, and luminal reduction of the proximally and distally adjacent cerebral vessel lumen was reported, but was rated to be clinically non-significant [25]. 

In this regard, it is important to investigate further whether device design and the extent of endothelial irritation associated with FDS oversizing achieve a synergistic vasospasmogenic effect. The impact of stent design on the development of neointimal hyperplasia, another predominantly flow-diverter-associated phenomenon inducing in-stent stenosis, was substantiated quite recently in a number of studies [7,26,27]. Hence, it seems comprehensible that acute and prolonged endothelial injury, which are evoked by the implantation of endovascular devices, initiate endothelial necrosis and a potentially critical, cell-biological cascade culminating in local vasospasm, neointimal hypertrophy, and thromboembolism [27,28,29].

In our collective, both conditions—subacute vasospasm and neointimal hyperplasia—occurred in an overlapping fashion to different extents. Most importantly, the singular case, which manifested as prolonged ischemic neurological deficit, exhibited severely stenosing proximal and distal segmental vasospasm aggravated by moderate neointimal hyperplasia. The vasospastic nature of the FDS deformation was evidenced by the strong immediate therapeutic effect of intra-arterial biochemical angioplasty. However, repetition of i.a. spasmolysis twice a day for a period of seven days was necessary to maintain sufficient perfusion of the depending brain parenchyma, as the segmental vasospasm relapsed repeatedly, peaking approximately 12 h after the last treatment. 

Interestingly, most patients with relevant segmental vasospasm apparent in early follow-up imaging complained of novel, migraine-like headaches, which appeared at the time of the empirical maximum of vasospasm, ipsilateral to side of the treated aneurysm. This phenomenon is observable in our cohort and certainly related to the reactive increase of blood flow via the middle meningeal artery, physiologically compensating for the decreased perfusion due to stenosis of the ICA or its major branches [30].

Our study suffers from a number of limitations. Firstly, it is composed of an inhomogeneous small patient collective, which was investigated with different follow-up strategies. Furthermore, the pre-selection of patients based on apparent segmental vasospasm represents a potentially considerable bias. Therefore, interpretation of our statistical results requires caution and corroboration by adequately designed subsequent investigations. Also, intermediate and long-term follow-ups are not yet available in our patient selection. The true importance of oversizing, radial outward force, porosity, device location within the cerebral vessels, and other variables on subacute vasospasm and neointimal hyperplasia can only be discussed in our report, as prospective studies including more patients from different centers are necessary to determine this conclusively.

### Outlook

Neurovascular centers performing FDS implantations should consider implementing a specific follow-up algorithm including radiography after 3–4 weeks to detect potentially critical subacute vasospasm (which may be further aggravated by additional neointimal hyperplasia) and standard DSA imaging three and nine months post implantation. Initial MRI prior to implantation and 6–9 months post procedure is also recommended to identify device-related delayed ischemia and allow for better risk stratification in further patients. Also, from our current experience, oversizing, especially at the ICA–MCA junction (and probably also at the vertebrobasilar confluens) should be strictly avoided. If the anatomy at hand increases the probability of shortening, an adequately longer device instead of a device with greater diameter should be considered. 

## 5. Conclusions

Segmental vasospasm appears to be a frequent, vascular reaction peaking 3–5 weeks after endovascular implantation of devices with high surface coverage. It occurs in patients who suffered from SAH and in patients with incidental, unruptured aneurysms. Distinct locations of the treated segments and oversizing of the FDS were associated with the occurrence of segmental vasospasm. As one patient with severe segmental vasospasm suffered minor stroke and required repeated intra-arterial spasmolysis to prevent a disabling or potentially critical outcome, we recommend establishing an early follow-up imaging to screen for patients at risk for stroke after FDS treatment. However, further studies on a prospectively collected, larger patient cohort are required to validate our findings and elucidate the phenomenon further. 

## Figures and Tables

**Figure 1 jcm-08-01649-f001:**
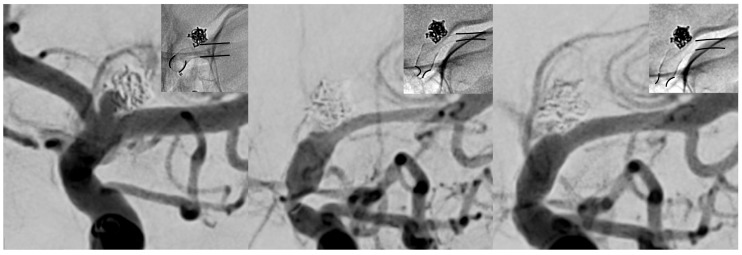
Subsequent imaging findings of the symptomatic patient from uneventful implantation (**left**), to clinically manifest vasospasm (middle, three weeks post procedure) and after the first session of intra-arterial (i.a.) treatment (**right**). Note the change in caliber of the proximal and distal landing zones (black lines in each radiogram underline the change of caliber of the implanted flow-diverting stent (FDS)). The left image demonstrates normal calibers of vessel and implanted FDS approximately 30 min after the procedure. The middle image reveals high-grade tandem stenosis of the terminal internal carotid artery (ICA) and the M1 segment, caused by subacute vasospasm resulting in severe stent compression. The right image shows a significant increase in caliber immediately after i.a. spasmolysis, accompanied by clinical recovery of the patient.

**Figure 2 jcm-08-01649-f002:**
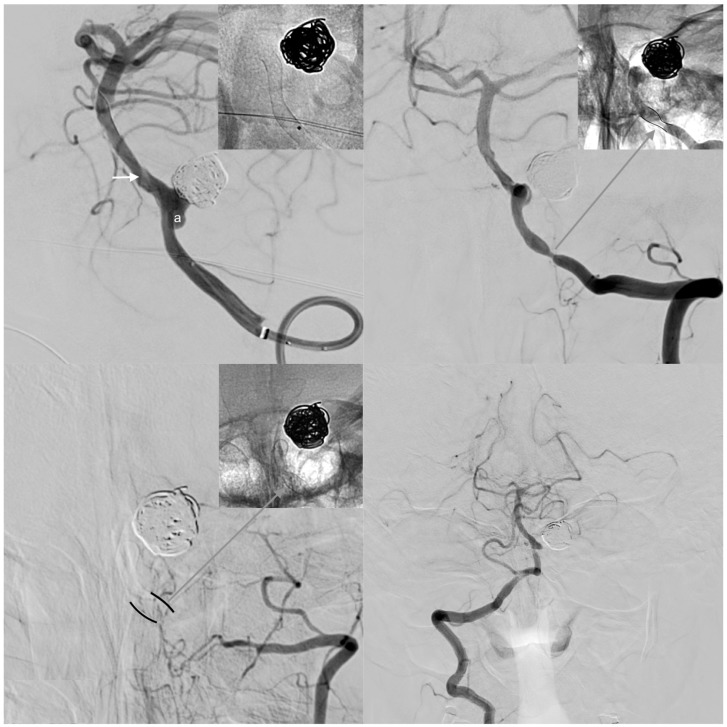
Clinically inapparent case of vasospasm-associated occlusion of a left-sided, dominant vertebral artery after FDS implantation for treatment of a de novo aneurysm arising from the posterior inferior cerebellar artery (PICA) orifice. Upper row: The left image shows the left V4 segment carrying a broad-based, de novo aneurysm (a) evolving in close proximity to a previously coiled PICA aneurysm. Note the radiogram showing the optimally implanted, well-unfolded FDS, which intentionally spared the VBA confluens aiming to preserve the hypoplastic right vertebral artery as a potentially important collateral vessel (white arrow: wash-out caused by inflow from the right-sided V4). The right image shows the early follow-up DSA five weeks after implantation. Note the short, high-grade stenosis along the proximal landing zone. The radiogram detail in the upper right corner illustrates compression of the stent (black lines), additional neointimal hyperplasia (white area bordering the stent), and the residual lumen (gray area). The patient complained of recurrent episodes of cervical–occipital headaches on the left side, but otherwise remained asymptomatic.Inferior row: control angiogram and non-enhanced device image five months after treatment. The left V4 segment is occluded, and the V3 segment is reduced in size. Note the re-unfolded FDS in the radiogram (upper right corner). The formerly hemodynamically non-essential right-sided vertebral artery now independently supplies the posterior fossa. The patient did not experience a neurological deficit at any time.

**Figure 3 jcm-08-01649-f003:**
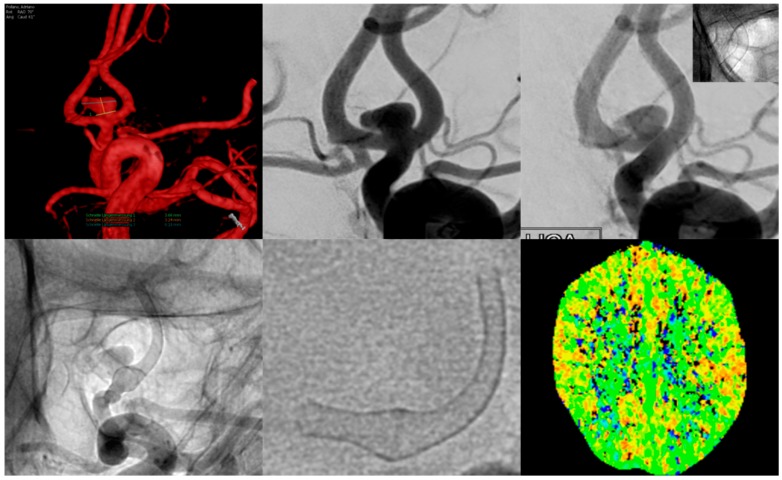
Example of a patient suffering from migraine-type left-sided headache for approximately two weeks, beginning three weeks after flow diverter implantation.The upper row shows a three-dimensional (3D) rotational angiogram (left) and the conventional working projection (middle) of the AcomA-complex with the aneurysm mainly being supplied by the left A1 segment. The right image shows normal calibers of the vessel and FDS after implantation.Inferior row: Conventional, non-subtracted angiogram (left) of the left ICA shows a moderate–high-grade tandem stenosis of the proximal and distal landing zones (A1 and A2 segment of the left ACA) caused by vasospasm. The middle image shows the FDS in free projection; note the normal caliber along the aneurysm-bearing segment and the marked narrowing of the proximal and distal landing zones. CT perfusion imaging (right image) revealed no significant decrease in perfusion of the left ACA territory.

**Figure 4 jcm-08-01649-f004:**
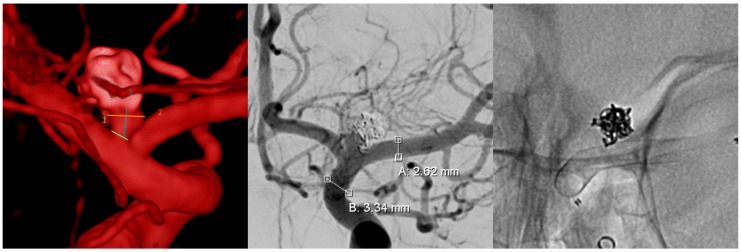
Digital subtraction angiography (DSA) images of the FDS implantation of the patient who experienced symptomatic vasospasm three weeks later. From left to right: three-dimensional (3D) rotational angiogram of the recurrent aneurysm, conventional posterior anterior (p.a.) angiogram in working projection, and corresponding radiogram of the uneventfully implanted FDS.

**Figure 5 jcm-08-01649-f005:**
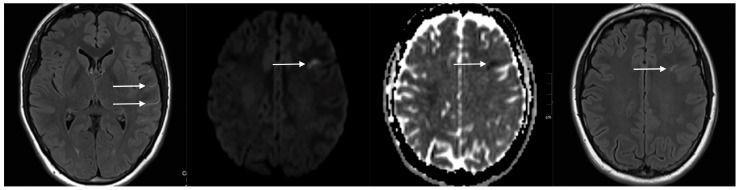
Representative magnetic resonance imaging (MRI) sections of the patient presenting with acute global aphasia three weeks after FDS implantation. From left to right: Fluid Attenuated Inversion Recovery (FLAIR) sequence reveals slow flow in the hyperintense M4 segments of the left hemisphere (white arrows). Diffusion weighted imaging (DWI) shows acute-stage subcortical infarction in the left-sided, frontal middle cerebral artery (MCA) territory. Corresponding FLAIR section confirms the subcortical infarction revealed by DWI.

**Figure 6 jcm-08-01649-f006:**
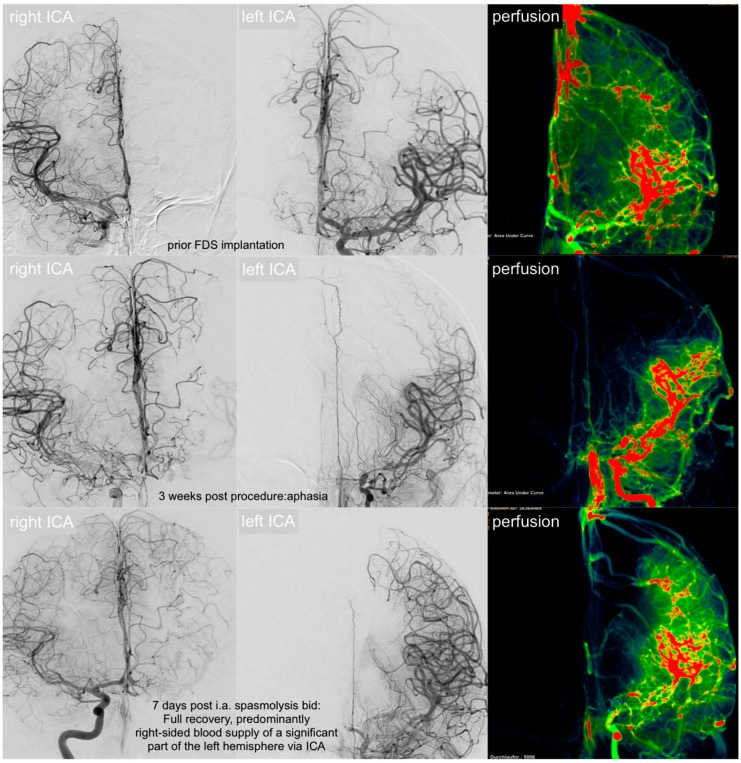
Changes in brain perfusion of the most severe vasospasm case, beginning with the unremarkable status after intervention (upper row), followed by the status when presenting with global aphasia and severe, unilateral headache (middle row, three weeks later), and after a week of mean arterial pressure-driven i.a. anti-vasospastic treatment (inferior row). Upper row—initial status: conventional p.a. angiogram of the right ICA, left ICA, and DSA perfusion prior to FDS implantation: functionally autonomic supply of both hemispheres via each ipsilateral ICA. Middle row—acute-phase vasospasm: significant collateral supply from the right ICA via the AcomA along left A1 for compensation of acute-onset, hemodynamically critical vasospastic stenosis. Note the severely reduced left-sided cerebral blood volume represented by area under the curve perfusion image. Inferior row—equilibrated collateral flow and increase in left-sided brain perfusion after multiple intra-arterial treatments for segmental vasospasm. Eventually, the left and right ACA territory is supplied from the right ICA. Left MCA perfusion is sufficient via the left ICA.

**Table 1 jcm-08-01649-t001:** Summary of clinical and technical features of all included patients.

Case	Sex	Age (years)	Pathology, Location, Strategy	Proximal Vessel Diameter in mm	Distal Vessel Diameter in mm	Implanted Device	Maximal Device Oversizing	Vasoconstricive Segmental Stenosis %; Location	Time Point of Follow-Up Imaging Post Implantation	Additional In-Stent Stenosis/Neointimal Hyperplasia	Cumulative Local Stenosis
1	Male	51	Left V4, ruptured dissecting aneurysm, primary Flow Diverting Stent (FDS)	4.2	2.5	PED 2 400-300	1.5 mm, 60%	15% distal landing zone (distal V4)	20 weeks	20%	35%
2	Male	72	Basilar artery, ruptured blister aneurysm, primary FDS	2.2	1.7	p48 HPC 300-18	1.3 mm, 56%	45% distal landing zone (basilar tip)	8 weeks	Radiography only, not assessable	Minimum of 30%
3	Female	51	Left M1, ruptured saccular aneurysm, Plug and Pipe (P&P)	2	1.4	P48200-15	0.6 mm, 30%	25% proximal landing zone (left M1)	4 weeks	Radiography only, not assessable	Minimum of 30%
4 *	Male	64	Left A1–A2, ruptured saccular aneurysm, P&P after pCONus 2 Stent-Assisted Coiling (SAC)	2.1	1.7	p48 HPC 300-18	1.3 mm, 43%	12% distal landing zone (A2; uncovered by pCONus)	5 weeks	Radiography only, not assessable	12%
5 *	Female	78	Right Middle Cerebral Artery (MCA) bifurcation, incidental saccular giant-aneurysm, elective P&P after pCONus 2 SAC	3.0	2.6	p48 HPC 300-18	0.4 mm, 13%	31% distal landing zone (right M2, not covered by pCONus)	5 weeks	Radiography only, not assessable	31%
6 #	Female	52	Basilar artery right P1, incidental saccular giant aneurysm, primary FDS	2.7	2.1	2 × p48 HPC 300-18	0. 9 mm, 30%	14% distal landing zone (right P1)	5 weeks	Radiography only, not assessable	14%
7	Male	33	Right A1, incidental saccular aneurysm, primary FDS	1.9	2.4	p48 HPC 300-15	1.1 mm, 42%	37% proximal landing zone (right A1)	5 weeks	Radiography only, not assessable	Minimum of 37%
8	Female	49	Left PcomA-ostium, ruptured saccular aneurysm, P&P	2.5	3.1	p64 300-15	0.5m, 20%	20% proximal landing zone (ICA)	14 weeks	10%	30%
9	Male	37	Right M1, incidental saccular aneurysm, elective P&P	2.6	2.4	PED2-275-12	0.35 mm, 15%	33% distal landing zone (M2) and 15% proximal landing zone (M1)	10 weeks	10%	43%
10	Male	51	Left A1–A2, incidental saccular aneurysm, primary FDS	1.8	1.8	SVB 2.25 × 15	0.45 mm, 25%	33% proximal landing zone (A1)	13 weeks	20%	53%
11	Female	18	Right M2, incidental saccular aneurysm, primary FDS	1.8	1.9	SVB 2.25 × 15	0.45 mm, 25%	19% proximal landing zone (M2)	12 weeks	22%	41%
12	Female	39	Left A1–A2,ruptured saccular aneurysm, P&P	1.8	1.8	SVB 2.25 × 20	0.45 mm, 25%	42% proximal landing zone (A1)	14 weeks	5%	47%
13	Female	50	Right A2–A3, incidental saccular aneurysm, primary FDS	1.8	1.7	SVB 2.25 × 10	0.55 mm, 32%	33% proximal landing zone (A2)	17 weeks	10%	43%
14 *	Female	38	Left A1–A2, incidental saccular aneurysm, primary FDS	1.7	2.0	SVB 2.25 × 10; 2.25 × 15	0.55 mm, 32%	18% proximal landing zone (A1)	28 weeks	13%	31 %
15	Female	55	Right A1/2, ruptured saccular aneurysm, P&P	1.7	1.6	SVB 2.25 × 15	0.65 mm, 40%	56% proximal landing zone (A1)	17 weeks	10%	66%
16	Female	48	Left PICA, ruptured saccular aneurysm, P&P	2.4	2.3	SVB 2.25 × 10	none	None	14 weeks	10%	10%
17 *	Male	55	Right A1–A2, incidental aneurysm, primary FDS	2.0	1.9	SVB 2.25 × 15	0.35 mm, 18%	33% proximal landing zones (each)	14 weeks	0%	33%
18	Female	40	A1–A2, ruptured saccular aneurysm, P&P	2.2	1.7	SVB 2.25 × 15	0.55 mm, 32%	33% of the proximal and distal landing zones (A1, A2)	9 weeks	10%	43%
19	Female	59	Left paraophthalmic Internal Carotid Artery (ICA), incidental saccular aneurysm, primary FDS	3.4	3.0	SVB 3.25 × 20	0.25 mm, 8%	17% distal landing zone (ICA)	12 weeks	0%	17%
20	Female	70	Right Posterior Communicating Artery (PcomA), incidental saccular aneurysm, primary FDS	3.1	3.0	SVB 3.25 × 25	none	None	12 weeks	0%	0%
21	Female	56	Left Posterior Inferior Cerebellar Artery (PICA), ruptured saccular aneurysm, P&P	3.4	3.2	SVB 3.25 × 10	0.05 mm, 1.5%	None	12 weeks	0%	0%
22	Female	39	Right A1–A2, ruptured saccular aneurysm, P&P	2.0	1.7	SVB 2.25 × 15	0.55 mm, 25%	25% proximal landing zone (A1)	16 weeks	10%	35%
23	Female	58	Left A1–A2, ruptured saccular aneurysm, P&P	2.3	2.1	SVB 2.25 × 15	0.1 mm, 7%	None	15 weeks	0%	0%
24 *	Female	58	Left RCP and left MCA, incidental saccular aneurysm, primary FDS	3.4	1.6	SVB 3.25 × 20; 3.25 × 25	1.6 mm, 51 %	Distal landing zone 25% M1	14 weeks	10%	35%
25 #	Female	64	Right A2–A3, saccular aneurysm, elective P&P after Leo-Baby + coiling	1.8	1.7 (covered by LEO baby stent)	SVB 2.25 × 10	0.55 mm, 32%	Proximal landing zone 13% (A2)	19 weeks	15%	28%
26	Male	38	Left ICA bifurcation + RCP, ruptured saccular aneurysm, P&P + primary FDS	3.6	2.5	SVB 3.25 × 20	0.75 mm, 30%	40% distal landing zone (M1) 30% middle–proximal quarter (ICA)	13 weeks	5%	45%
27	Female	27	Left M1, ruptured saccular aneurysm, P&P	3.5	2.8	SVB 3.25 × 20	0.45 mm, 16%	>85% of the proximal (ICA) and >60% of the distal landing zone (M1)	3 weeks	10%	>95%
28	Male	63	Left A1–A2, incidental saccular aneurysm, primary FDS	2.2	1.8	SVB 2.25 × 15	0.45 mm, 25%	38% of the distal (A2) and 27% of the proximal landing zone (A1)	16 weeks	0%	38%
29	Male	52	Left supraophthalmic ICA, ruptured saccular aneurysm, P&P	3.2	2.9	SVB 3.25 × 25	0.35 mm, 12%	7% distal landing zone	8 weeks	0%	7%
30	Male	40	Left PICA, ruptured saccular aneurysm, P&P	2.5	2.5	SVB 2.75 × 25	0.25 mm, 10%	42% proximal landing zone	6 weeks	67%	81%
31 *	Male	60	Right M1, ruptured saccular aneurysm, P&P	3.1	2.8	SVB 2.25 × 15; 2.75 × 15	none	0%	5 weeks	0%	0%
32	Male	54	Left A1–A2, ruptured saccular aneurysm, P&P	2.5	1.8	SVB 2.75 × 20	0,95 mm, 52%	48% proximal landing zone (A1) 44% distal landing zone (A2)	6 weeks	Radiography only, not assessable	48%
33	Female	55	Right A1–A2, ruptured saccular aneurysm, P&P	2.1	1.5	SVB 2.25 × 20	0,75 mm, 50%	33% distal landing zone (A2) 20% proximal landing zone (A1)	7 weeks	Radiography only, not assessable	33%
34	Female	61	Right A1–A2, ruptured saccular aneurysm, P&P	2.0	1.9	SVB 2.25 × 15	0,35 mm, 15%	50% distal landing zone 20% proximal landing zone	5 weeks	Radiography only, not assessable	50%
35	Male	35	Left A1–A2, incidental saccular aneurysm, primary FDS	3.0	2.5	p48 3 × 18	0.5 mm, 20%	53% distal landing zone, 40% proximal landing zone	4 weeks	0%	53%
36	Male	36	Left V4, incidental saccular aneurysm, primary FDS	3.1	2.5	SVB 2.75 × 25	0.25 mm, 10%	72% proximal landing zone	5 weeks	10%	82%

* Patients treated with FDS implantation within a previously implanted aneurysm stent. # Patients treated with 2 FDS simultaneously–telescoping.

**Table 2 jcm-08-01649-t002:** Summary of results after grouping our patients according to their individual degree of vasospastic stenosis.

Vasospastic Stenosis in %	Number of Patients	Average Oversizing in % (Mean ± SD)	Average Resulting Device Compression in % (Mean ± SD)	Female/Male Patients	Average Age in Years	Average Size of the Aneurysm (Mean ± SD)
0–24	14	19.3 ± 17.8	9.6 ± 7.8	10/4	52.8	7.5 ± 4.8
25–50	16	31.3 ± 14.3	34.8 ± 6.7	7/9	50.9	5.7 ± 6.0
>50	6	21.8 ± 10.2	59.3 ± 14.9	3/3	42.0	5.6 ± 1.4

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
