# Peer review of "Delayed Stroke after Aneurysm Treatment with Flow Diverters in Small Cerebral Vessels: A Potentially Critical Complication Caused by Subacute Vasospasm"

_jcm, 2019, doi:10.3390/jcm8101649_

Round 1

Reviewer 1 Report

In this study authors describe vasospasm phenomena after FDS implantation in 36 patients. While this phenomena is novel and can have relevance for clinical practice but there are numerous shortcomings in the report:

It remains not exactly clear what number of patients received early follow-up. In the abstract authors state that 36, while in the Methods section they state that in 36 patients "suitable FU had been performed". Authors considered only gender, aneurysm location and endovascular approach. I believe that important clinical factors ruptured vs. unruptured status, disease severity (GCS, HH grade, Fisher grade etc) should be considered as they can substantially modify rupture risk.  Was nimodipine used in SAH patients? I would also recommended including other biochemical and CBC etc. variables to better describe this novel observation. Was aneurysm size considered?  I would recommended not using FU as abbreviation and spell out follow-up for clarity purposes.  What were the underlying causes for selection of different FDS devices? Were there any differences as function of device? Given this observation, do authors recommend earlier testing for vasospasm after FDS implantation? What could be done to prevent from it? Are preventive therapies indicative?

Author Response

Dear reviewer 1, thank you very much for the time and effort you invested in our manuscript and your valuable comments. Language and grammar were revised with the help of a native speaker. Title and conclusions were amended according your input. Furthermore, all comments were addressed as follows:

The method section regarding the follow-up period has been corrected to increase comprehensibility according your comment.

As suggested, the requested factors were also included in the analysis - ruptured vs. unruptured aneurysm status, disease severity according Hunt & Hess as well as Fisher’s scale and size of the respective aneurysms. The manuscript was amended accordingly. In brief, neither the clinical nor the morphological groups reflecting the severity of acute SAH were related to the occurrence of vasospasm or neointimal hyperplasia in our patients. Also, aneurysm size was not associated with the severity of delayed, FDS induced vasospasm.However, as the number of included patients is small and the collective is relatively heterogeneous, complementary studies are necessary to validate the findings of our pilot study. Considering previous reports on aneurysm size, risk of rupture and outcome (AlMatter et al., 2019 “The size of ruptured intracranial aneurysms, Clin Neuroradiol; Salary et al., 2007 “Relation among aneurysm size, amount of subarachnoid blood, and clinical outcome”, J Neurosurg; Bhogal et al., 2018 “Difference in aneurysm characteristics between ruptured and unruptured aneurysms in patients with multiple intracranial aneurysms”, Surg Neurol Int.), we hypothesize that the size of the aneurysm is not related to the onset or degree of device induced vasospasm.

FU is no longer abbreviated in the revised manuscript, as recommended.

Nimodipine is used as a standard component in all coiling or stent-assisted coiling procedures in our institution. In case of flow diverter implantations we do not use nimodipine on a regular basis, as it indirectly may result in unintended oversizing – which we believe may induce delayed vasospasm. However, in case segmental vasospasm occurs during the intervention, nimodipin or nitrolingual are used individually to resolve vasospasm related to endovascular manipulations.

Thank you for this valuable suggestion. We are currently conducting a prospective analysis of standard laboratory data and blood-derived biomarkers from patients presenting with subacute, post SAH-vasospasm or delayed, device induced vasospasm together with immune-histochemical investigations of neuropahological specimens focusing on endothelial signaling related to the induction of segmental vasoconstriction and neo-angiogenesis. In our opinion, this study will provide relevant data in this regard, however, related to the patient collective, extent and distinct nature of the investigation it will be presented in a separate manuscript.

The choice of the individual device was mainly based on the following considerations: a) accessibility of the target  segment / navigational features of the delivery micro-catheter, b) empirical extent of the flow diverting effect of the individual devices in relation to the assumable number of perforating or eloquent major branches of the target segment,  c) availability of the individual device in the dimensions matching the target segment and d) actual availability of the devices. Conclusively, each device was chosen after careful consideration of the individual hemodynamic and anatomical situation at hand in combination with the availability of a suitably sized device.

First of all we recommend to implement an adapted follow up regimen to identify patients at risk for stroke related to the occurrence of delayed vasospasm. As reported, the culmination of vasospasm after FDS implantation occurs at 3-5 weeks after the procedure. Therefore we strongly recommend to perform a simple radiogram of the flow diverter ca. 4 weeks post implantation for screening purposes. In clinically asymptomatic patients - if severe deformation of the device is apparent, DSA is recommended to assess the perfusion of the dependent parenchyma and the potentially available compensatory collateral flow. If perfusion is reduced critically, we recommend to admit the patient to a dedicated IMC/ICU for continuous observation and euvolemic hypertension.
In clinically affected patients, more specifically in patients exhibiting a distinct focal neurological deficit, immediate DSA is recommended to assess the cerebrovascular status of the patient and the potential need for i.a. therapy.
As also included in our manuscript, moderate oversizing apparently results in the provocation of delayed vasospasm and certain locations are seemingly more susceptible for the phenomenon, for example the ICA-MCA junction and the VBA confluens. As implanting a flow diverter along the ICA-MCA junction or covering the vertebral artery confluens certainly compromises potentially essential collateral flow from the contralateral side, we strongly advise to include this aspect in treatment planning and  if possible avoid such scenarios. In this context please review figure 1: flow diversion at the ICA-MCA junction provoking ICA and M1-spasm can prevent collateralization via the AcomA - and figure 2: flow diversion covering the vertebral artery confluens may jeopardize potentially essential collateral flow supplying the basilar territory via the contralateral vertebral artery.

So far, no predictive testing for cerebral vasopasm after FDS implantation is available. In cardiology, provocative testing for coronary reactivity and spasm is a well established method (for example Zaya et al., 2014 ‘Provocative testing for coronary reactivity and spasm’, JACC). Considering this, future investigations on the safety and value of this method for neuroendovascular procedures may be helpful to identify patients susceptible for device induced segmental vasospasm.

Again - we want to thank you for our help to improve our manuscript. We look forward to any further questions that you may have

Reviewer 2 Report

Shob and coauthors have reported advantage of novel endovascular techniques Flow-Diversion (FD) for identification of patients with risk for ischemia and potential anti-vasospasm therapy. Their results showed subacute, device-induced vasospasm as frequent cause for critical perfusion reduction in patients with FDs.

This is excellent study on important topic. The study is very well written, and results are extensive discussed. The rigorous set up was implied. The study brings important new information about advantages and disadvantages of new hemodynamic treatment of cerebral aneurism by implying flow diverting stents.  There are not any concerns.

Author Response

Dear  reviewer 2, thank you very much for your encouraging review. We appreciate the time and effort you invested in our manuscript. 

Reviewer 3 Report

Schob and coworkers evaluated the occurrence of segmental vasospasm in patients treated with flow diverters. They found that this phenomenon can potentially lead to critical complications including delayed stroke.

Essentially, this manuscript is a collection of several cases, based on a pre-selection of unconventionally early follow-up less than 6 months after initial treatment. The cohort as is is therefore biased and this has implications on the remainder of the study.

Besides the fact that the power of the current study is limited due to the limited number of patients, the statistical analyses and their validity are disputable, as this population is already biased by the pre-selection based on segmental vasospasm. Without knowledge on overall population conclusions on discriminatory value of gender etc. might be far-fetched.

Table 1: please also show summarized data, either as means and/or as categories to aid reader in understanding overall patient characteristics.

How was stenosis quantified? Please mention in methods.

Please define biochemical angioplasty for the non-expert reader.

Author Response

Dear reviewer 3, thank you very much for all the time and effort which you invested in our manuscript including your valuable comments. Your comments were addressed as follows:

Thank you for this valuable comment – the aspect of the pre-selection bias has been included in the discussion section of the revised manuscript.

An additional table (2) providing a summarized overview over our results has been created and included according your suggestion.

Thank you for this helpful comment. The method section was amended regarding accordingly.

Biochemical angioplasty is now explained in the revised version of the manuscript.

We again want to express our gratitude for your help to enhance our manuscript. We are looking forward to answering any further questions that you may have.

Round 2

Reviewer 3 Report

The authors have addressed my concerns. The biased selection still poses a problem but I do not see any way the authors can solve this at this time.